# Determination of Critical Phosphorus Dilution Curve Based on Capsule Dry Matter for Flax in Northwest China

Yaping Xie [1], Yang Li [2], Limin Wang [1], Mir Muhammad Nizamani [3], Zhongcheng Lv [4], Zhao Dang [1], Wenjuan Li [1], Yanni Qi [1], Wei Zhao [1], Jianping Zhang [1,*], Zhengjun Cui [5], Xingrong Wang [1], Yanjun Zhang [1] and Gang Wu [6]

1   Crop Research Institute, Gansu Academy of Agricultural Sciences (GASS), Lanzhou 730070, China
2   College of Agronomy, Shanxi Agricultural University (SXAU), Taiyuan 030006, China
3   School of Life and Pharmaceutical Sciences, Hainan University (HNU), Haikou 570228, China
4   Ordos Institute of Agricultural Sciences (OIAS), Dongsheng 017000, China
5   College of Agronomy, Gansu Agricultural University (GSAU), Lanzhou 730070, China
6   Zhangye Water-Saving Agricultural Experimental Station, Gansu Academy of Agricultural Sciences (GAAS), Zhangye 734000, China
*   Correspondence: zhangjpzw3@gsagr.ac.cn; Tel.: +86-0931-761-108-1

**Abstract:** One of the cores of flax (*Linum usitatissimum* L.) production is to precisely measure the requirement of phosphorus (P) fertilization for optimizing seed yield, grower profits, P-use efficiency, and reducing environmental risk. Therefore, critical P concentration (Pc) was proposed as a suitable analytical tool to assess the flax P nutrition status. Four field experiments, with five P applications (0, 40, 80, 120, 160 kg $P_2O_5$ ha$^{-1}$) and four cultivars (Longyaza 1, Longya 14, Lunxuan 2, and Dingya 22) were conducted from the 2017 to 2019 seasons. The capsule Pc dilution curve based on capsule dry matter (CDM) was described by Pc = 2.84 × CDM$^{-0.22}$ ($R^2$ = 0.87, $p$ < 0.01), CDM ranging from 0.60 to 4.17 t ha$^{-1}$. The P nutrition index (PNI) exhibited a significant positive relationship with P application rate. In addition, the relative seed yield was closely related to PNI. Those results validate that the capsule Pc dilution curve can be an alternative and more rapid tool to diagnose flax P status to support P fertilization precise decisions during the reproductive growth of flax in northwest China.

**Keywords:** phosphorus dilution curve; flax; phosphorus nutrition index; relative seed yield; phosphorus nutrition status

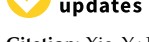



## 1. Introduction

There has been a growing interest in flax (*Linum usitatissimum* L.), known as oilseed flax or linseed [1], because industries have requested increased quantities of seeds in China. At present, available arable land is declining, so maximizing crop yield per unit area has become a significant aim of agricultural production in China [2,3]. Previous several studies have reported that phosphorus (P) fertilization improved the production of many oil crops, such as flax [4–6], canola [7], safflower and sunflower [8], soybean [9], etc. However, excessive P fertilization in crop production happened occasionally, which resulted in a series of concerns for environmental, ecological, and human health [10–12]. Therefore, the precise management of P fertilization of flax has become a core area of research.

Zamuner et al. [13] indicated that an ideal indicator of a crop P's nutritional status should show P deficiencies and excesses, provide rapid diagnosis, and allow correction during the growing season. In this case, critical P concentration (Pc) was developed, which is defined as the minimum plant P concentration needed to achieve maximum crop biomass, which is a suitable analytical tool to assess the crop P nutrition status [14–16]. Curves of Pc have been generated for many crops, such as potato [13,17], wheat [14,15], timothy [18], mungbean and urdbean [19], rapeseed and maize [15]. Moreover, Pc dilution curves vary among different regions, species, genotypes within species, and practice management [20].

The crop yield and quality are associated with P nutrient status. The Pc dilution curve could be useful as a reference curve to assess flax's nutritional status through the P nutrition index (PNI). Analogously to the N nutrition index, the PNI can be calculated as the ratio between actual plant P concentration and the Pc expected according to that actual crop biomass [14–17].

For diagnosing the P status and estimating the appropriate P fertilizer requirements of flax during the reproductive growth period, it is essential to develop a Pc dilution curve based on the capsule dry matter (CDM) of flax in northwest China. Furthermore, the quantitative assessment of seed yield in response to PNI is highly required to validate the Pc dilution curves as a robust diagnostic tool in flax production. Therefore, the objectives of this study were to establish and validate a Pc dilution curve based on CDM, to assess the relationship between relative seed yield (RY) and PNI in response to flax under different P rates in northwest China for improving P-use efficiency and environmental protection of flax production.

## 2. Materials and Methods

### 2.1. Experimental Site

Field experiments 1 and 2 were conducted from 2017 to 2018 at Dingxi Academy of Agricultural Science (34.26° N, 103.52° E, altitude of 2060 m) in Gansu, China, as described in detail in Table 1. Experiments 3 and 4 were carried out from 2018 to 2019 at Yongdeng, Gansu, China (36°02′ N, 103°40′ E, and altitude 2149 m). The soil type is classified as Arenosols [21]. The previous crops for the four experiments were all wheat.

**Table 1.** Basic information at the Dingxi site in the 2017 and 2018 growing seasons.

| Growing Season | Soil Characteristics | Cultivar | P Rate (kg $P_2O_5$ ha$^{-1}$) | Sampling Data |
|---|---|---|---|---|
| 2017 (Exp 1) | Type: loam Organic matter: 10.2 g kg$^{-1}$ Total N: 0.98 g kg$^{-1}$ Available P: 11.7 mg kg$^{-1}$ Available K: 122.5 mg kg$^{-1}$ pH: 7.9 | Lunxuan 2 Dingya 22 | 0 ($P_0$) 40 ($P_{40}$) 80 ($P_{80}$) 120 ($P_{120}$) | 86 92 98 104 110 |
| 2018 (Exp 2) | Type: loam Organic matter: 11.0 g kg$^{-1}$ Total N: 1.01 g kg$^{-1}$ Available P: 12.6 mg kg$^{-1}$ Available K: 135.4 mg kg$^{-1}$ pH: 8.1 | Lunxuan 2 Dingya 22 | 0 ($P_0$) 40 ($P_{40}$) 80 ($P_{80}$) 120 ($P_{120}$) | 86 92 98 104 110 |

Exp 1 = experiment 1; Exp 2 = experiment 2; N = nitrogen; P = phosphorus; K = potassium.

### 2.2. Experimental Design

Data were obtained from four field experiments in which the P rates, flax cultivars, sites, the physicochemical property of pre-planting soil, and varied years were summarized in Tables 1 and 2.

A randomized complete block design with three replicates was used for this study with a plot size of 20 m$^2$ (4 m × 5 m). Four P application rates (0, 40, 80, and 120 kg $P_2O_5$ ha$^{-1}$) were applied to two flax cultivars (Table 1), and five P rates (0, 40, 80, 120, and 160 kg $P_2O_5$ ha$^{-1}$) were applied to two other flax cultivars (Table 2). The cultivars were mainly planted in the local agriculture department and farms. Phosphorus fertilizer was applied using calcium superphosphate and broadcast uniformly over the soil surface before seedbed preparation and sowing. Before seedbed preparation and sowing in each site year, 80 kg N ha$^{-1}$ and 120 kg $K_2O$ ha$^{-1}$ were broadcast uniformly over the soil surface using urea and potassium sulfate, respectively. Forty kg N ha$^{-1}$ of urea was top-dressed at the budding stage. To ensure the maximum potential productivity, 40 mm of water was used

to irrigate each plot before the flowering of flax. Further crop management procedures followed common agricultural practices to ensure the maximum potential productivity, i.e., no factor other than P was limiting.

**Table 2.** Basic information at the Yongdeng site in the 2018 and 2019 growing seasons.

| Growing Season | Soil Characteristics | Cultivar | P Rate (kg $P_2O_5$ ha$^{-1}$) | Sampling Data |
|---|---|---|---|---|
| 2018 (Exp 3) | Type: Arenosols Organic matter: 9.8 g kg$^{-1}$ Total N: 1.23 g kg$^{-1}$ Available P: 10.0 mg kg$^{-1}$ Available K: 178.3 mg kg$^{-1}$ pH: 7.5 | Longyaza 1 Longya 14 | 0 ($P_0$) 40 ($P_{40}$) 80 ($P_{80}$) 120 ($P_{120}$) 160 ($P_{160}$) | 86 92 98 104 110 |
| 2019 (Exp 4) | Type: Arenosols Organic matter: 7.6 g kg$^{-1}$ Total N: 1.05 g kg$^{-1}$ Available P: 8.7 mg kg$^{-1}$ Available K: 141.6 mg kg$^{-1}$ pH: 8.2 | Longyaza 1 Longya 14 | 0 ($P_0$) 40 ($P_{40}$) 80 ($P_{80}$) 120 ($P_{120}$) 160 ($P_{160}$) | 86 92 98 104 110 |

Exp 3 = experiment 3; Exp 4 = experiment 4; N = nitrogen; P = phosphorus; K = potassium.

### 2.3. Sampling and Measurement

The 30 plants of flax per plot (2 by 1 m) were manually harvested during the reproductive stage (days after sowing 86, 92, 98, 104, and 110 days). Flax capsules (Figure 1) were collected and dried at 75 °C until a constant weight each date. Dried capsules were ground in a sample mill, passed through a 1 mm sieve, and samples were digested using $H_2SO_4$-$H_2O_2$, after which the CPC was determined by the Colorimetric Molybdenum-Blue method according to Lithourgidis et al. [22].

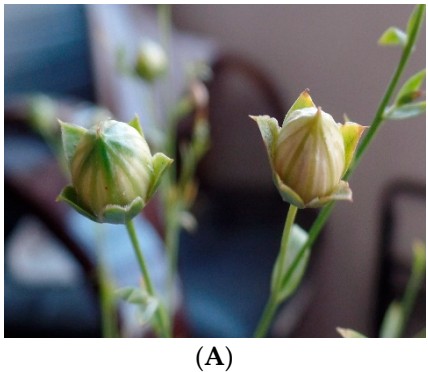
(**A**)

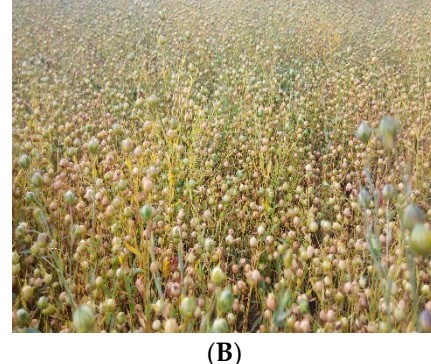
(**B**)

**Figure 1.** The capsules of flax (**A**) at 98 days and capsules of flax (**B**) at 104 days, respectively.

On the harvest date, the crop in each plot was separately harvested using a sickle to determine the seed yield.

### 2.4. Data Analysis
#### 2.4.1. Construction of the Pc Dilution Curve

The construction of a capsule Pc curve requires identifying critical data points at which the P neither limits growth nor enhances it. The data was collected to determine the Pc dilution curve during the 2018 and 2019 growing seasons at Yongdeng. A P-limiting treatment was defined as a treatment in which additional P led to a significant increase in CDM. A non-P-limiting treatment was defined as one in which P application did not lead to an increase in CDM.

The CDM and CPC values with different P rates were compared using ANOVA (SPSS 19 Software, Inc., Chicago, IL, USA) at a probability level of 5%, and treatment effects were determined using the least significant difference (LSD). A power regression equation was fitted to these theoretical, critical points to determine the equation of the Pc dilution curve. The Pc curve can be established by the following power equation [18]:

$$p_c = \alpha \mathcal{W}^{-b} \tag{1}$$

where *Pc* is the critical P concentration (g kg$^{-1}$); W is the total dry matter expressed in t ha$^{-1}$; *a* and *b* are positive constants, where *a* represents the critical plant P concentration in the dry matter (DM) when W = 1 t ha$^{-1}$ and *b* is a statistical parameter that represents the ratio between the relative decline in plant P concentration and the relative crop growth rate [20].

### 2.4.2. Phosphorus Nutrition Index

The P nutrition index (PNI) of the capsule in flax, used to characterize crop P status, was calculated according to the following formula [17,20]:

$$\text{PNI} = \frac{p_\alpha}{p_c} \tag{2}$$

where *Pa* was the actual capsule P concentration and *Pc* was the expected capsule P concentration. When PNI = 1, P nutrition is considered optimal. When PNI > 1, P nutrition is considered excessive; when PNI < 1, P nutrition is considered insufficient [12].

### 2.4.3. Relative Yield

The relative seed yield (RY) was obtained by dividing the seed yield at a given P rate by the highest seed yield among all P treatments [13,17]. The RY was calculated as the following equation:

$$\text{RY} = \frac{y_p}{y_h} \tag{3}$$

where *Yp* is the seed yield of flax in the fertilized plot, and *Yh* is the seed yield of the highest seed yield treatment (kg ha$^{-1}$).

The coefficients of determination ($R^2$) for the relationship between RY and PNI were calculated using SPSS 20.0.

### 2.4.4. Model Validation

In the current study, the root-mean-squared error (RMSE) and the normalized root-mean-squared error (n-RMSE) in 1:1 plots were used to evaluate the accuracy of the model, which is a common method used to identify the fitness of observed and estimated values [23]. When the n-RMSE $\leq$ 15%, it was looked at as a "good" agreement, 15–30% as a "moderate" agreement, and $\geq$30% as a "poor" agreement [24]. The RMSE and n-RMSE were calculated using Equations (4) and (5):

$$\text{RMSE} = \sqrt{\frac{\sum_{i=1}^{n}(S_i - m_i)^2}{n}} \tag{4}$$

$$\text{n} - \text{RMSE} = \frac{RMSE}{\bar{s}} \tag{5}$$

where n is the number of samples, $s_i$ is an estimated model value; $m_i$ is an observed value, and $\bar{s}$ is the averaged observed value.

### 3. Results

*3.1. Capsule Dry Matter and Capsule P Concentration at Different P Levels*

The P application significantly affected the CDM during the reproductive growth period. CDM increased gradually with the increase in P rate, with two years of value CDM averaging from 0.85 to 4.03 t ha$^{-1}$ and from 0.64 to 3.63 t ha$^{-1}$ in 2018 and 2019, respectively. In general, CDM increased significantly from $P_0$ to $P_{80}$ treatments. Still, there were no significant differences among $P_{80}$, $P_{120}$, and $P_{160}$ treatments. The maximum CDM was obtained in the $P_{80}$ treatment for two seasons and two cultivars (Figure 2). During each season, the CDM accorded the following inequality under different P levels:

$$CDM_0 < CDM_{40} < CDM_{80} = CDM_{120} = CDM_{160} \tag{6}$$

where $CDM_0$, $CDM_{40}$, $CDM_{80}$, $CDM_{120}$, and $CDM_{160}$ present the CDM value of $P_0$, $P_{40}$, $P_{80}$, $P_{120}$ and $P_{160}$, respectively.

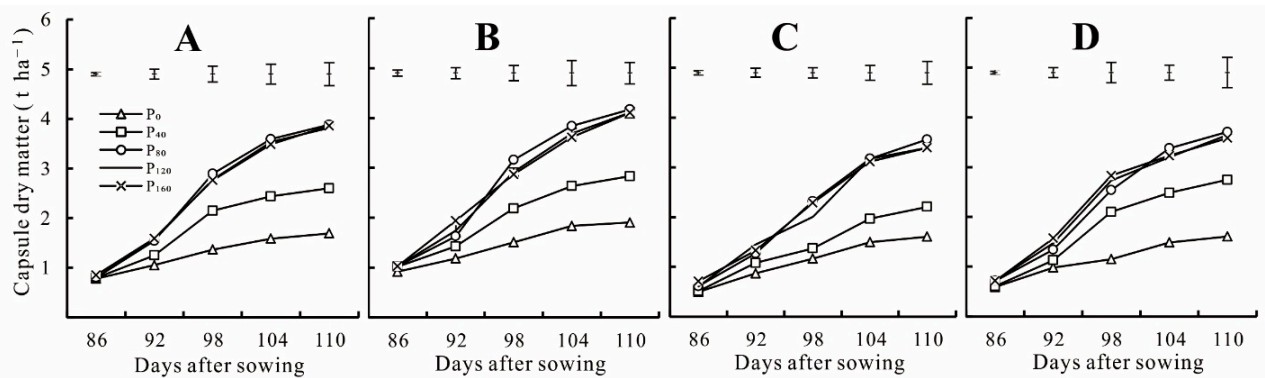

**Figure 2.** Changes in capsule dry matter for two flax cultivars at five P levels in two growing seasons. (**A**) Lonyaza 1 in 2018; (**B**) Longya 14 in 2018; (**C**) Lonyaza 1 in 2019 and (**D**) Longya 14 in 2019, respectively. The vertical bars indicate the least significant differences (LSDs) with $p \leq 0.05$ among five P levels (n = 3).

The CPC decreased gradually during the reproductive growth period of flax. A higher level of P generally resulted in a higher CPC (Figure 3). The CPC varied between 1.28 g kg$^{-1}$ DM and 3.71 g kg$^{-1}$ DM and from 1.08 g kg$^{-1}$ DM to 3.53 g kg$^{-1}$ DM in 2018 and 2019 across the two cultivars, respectively. The maximum CPC was obtained at the $P_{160}$ treatment for two cultivars and two growing seasons (Figure 3).

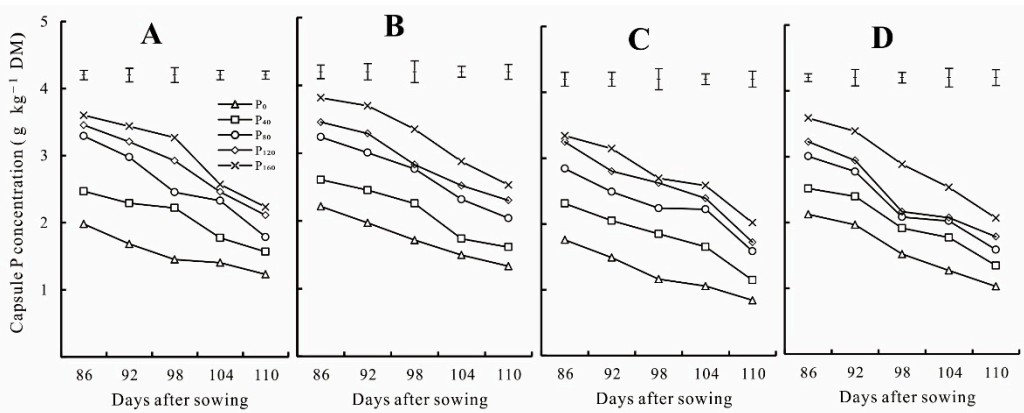

**Figure 3.** Changes in phosphorus concentration of the capsule for two flax cultivars at five P levels in two growing seasons. (**A**) Lonyaza 1 in 2018; (**B**) Longya 14 in 2018; (**C**) Lonyaza 1 in 2019 and (**D**) Longya 14 in 2019, respectively. The vertical bars indicate the least significant differences (LSDs) with $p \leq 0.05$ among five P levels (n = 3).

### 3.2. Constructing the Capsule Pc Dilution Curve for Flax

In this study, following the computational procedures of Justes et al. [25], capsule Pc points were established for each sampling date during the reproductive growth period in 2018 and 2019. The theoretical Pc points of CDM were decided for each sampling date, from capsule original to maturity for two cultivars, with 10 data points in the 2018 and 2019 seasons. A reducing trend of Pc points was observed with increasing CDM of flax (Figure 4).

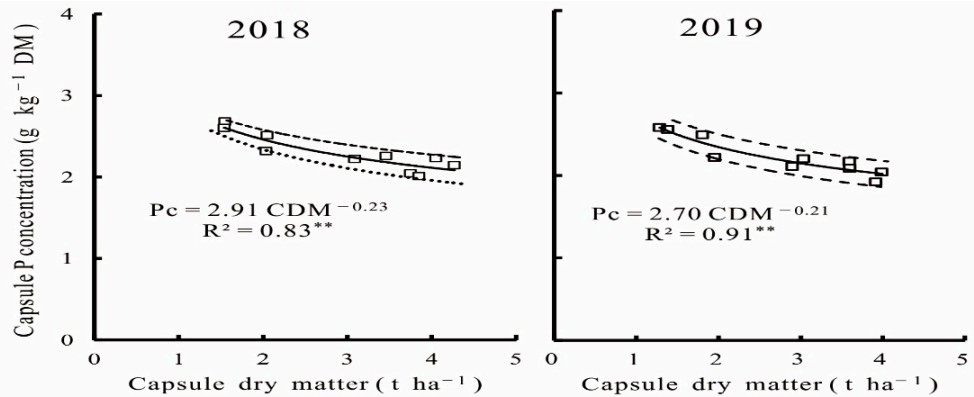

**Figure 4.** Critical phosphorus (Pc) data points were used to define the critical P dilution curve. The solid line represents the critical P dilution curve, depicting the relationship between the critical P concentration and the capsule dry matter of flax in Gansu, northwest China. The dotted lines represent the confidence intervals ($p = 0.95$). ** Significance at $p < 0.01$ probability level.

The following power equation could match the trend lines:

$$2018: Pc = 2.91\ CDM^{-0.23} \tag{7}$$

$$2019: Pc = 2.70\ CDM^{-0.21} \tag{8}$$

Two years expressed non-significant differences when compared according to calculation procedures recommended by Mead and Curnow [26]. Therefore, data points from two years were pooled to develop the following unified Pc curve (Figure 5):

$$Pc = 2.84\ CDM^{-0.22} \tag{9}$$

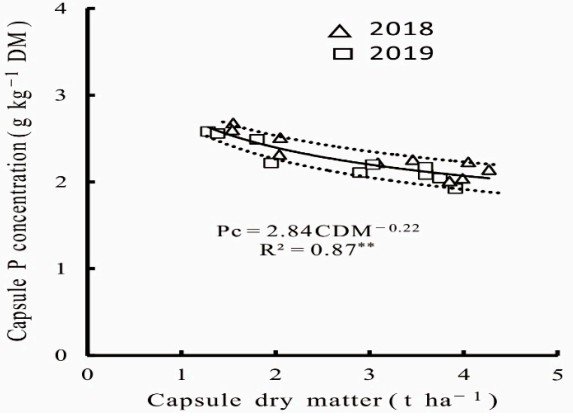

**Figure 5.** Critical phosphorus (Pc) data points for the capsule Pc curve definition using pooled data from two years. The solid line represents the critical P dilution curve (Pc = 2.84 CDM$^{-0.22}$, $R^2 = 0.82$ **), which describes the relationship between the critical P concentration and the capsule dry matter of flax in northwest China. The dotted lines represent the confidence intervals ($p = 0.95$). ** Significance at $p < 0.01$ probability level.

### 3.3. Validation of the Capsule Critical Phosphorus Dilution Curve

The capsule Pc curve was validated by an independent data set from the experiments conducted in 2017–2018 (Experiments 1 and 2, Table 1). The results expressed that the newly established curve can discriminate the P-limiting and non-P-limiting growth conditions of flax during the reproductive growth period. The curve was not affected by cultivar, season, and site. The data points of the P-limiting treatments were below or close to the Pc curve, while the points of the non-P-limiting treatments were near or above this curve (Figure 6). The accuracy of this model was assessed by using the RMSE and n-RMSE, using Equations (4) and (5). Results found that the RMSE of the model was 0.32 g kg$^{-1}$, and the n-RMSE was 13.86%, indicating "good" agreement between the observed and assessed values (Figure 6). Therefore, in the present study, we constructed that the CDM-Pc model can be used for the diagnosis of plant P nutrition.

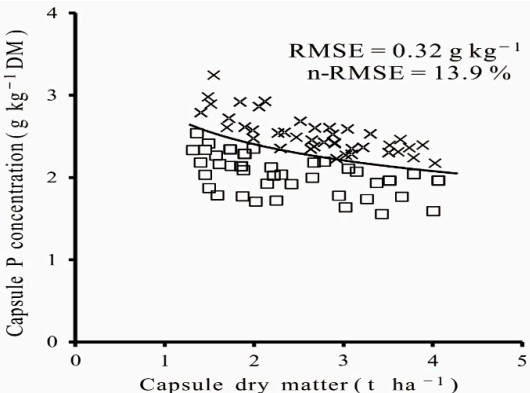

**Figure 6.** Validation of the critical phosphorus (Pc) curve using the independent data set from Experiments 1 and 2. Data points (□) and (×) represent P-limiting and non-P-limiting treatments, respectively. The solid line depicts the Pc curve based on the capsule dry matter of flax.

### 3.4. Change of Phosphorus Nutrition Index under Different P Treatments

The P nutrition index (PNI) is useful for diagnosing the crop P nutrition status. The PNI increased with an increasing P rate, ranging from 0.49 to 1.35 (Figure 7). For two years, the PNI values were <1 for the $P_0$ and $P_{40}$ treatments, indicating the two levels were insufficient for P nutrition. While the values of PNI were >1 for the $P_{120}$ and $P_{160}$ treatments, indicating the presence of a surplus of P uptake in the capsule of flax. The values of PNI were close to 1 for $P_{80}$, and this indicates that the P rate is optimal for P nutrition for flax growth. Therefore, the optimal P rate was 80 kg $P_5O_2$ ha$^{-1}$. These results elucidate that PNI can be used as a robust diagnostic tool for the P status of flax under different P conditions.

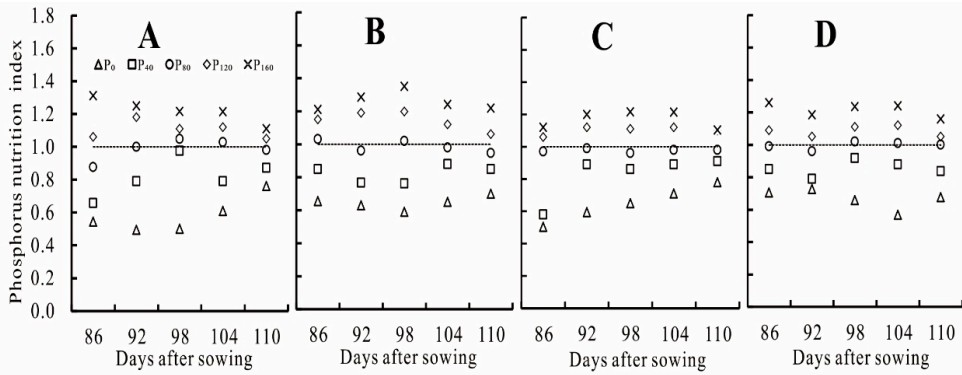

**Figure 7.** Phosphorus nutrition index (PNI) of two flax cultivars with five P rates in two years. (**A**) Lonyaza 1 in 2018; (**B**) Longya 14 in 2018; (**C**) Lonyaza 1 in 2019 and (**D**) Longya 14 in 2019, respectively. The reference line at PNI = 1 represents optimal P nutrition, while PNI > 1 shows excess P fertilizer application, and PNI < 1 shows P deficiency.

### 3.5. Relationship between Relative Seen Yield and PNI

The relationship between relative RY and PNI was well illustrated with a second-order polynomial equation (RY = –1.75 PNI² + 3.63 PNI–0.88, R² = 0.92 **). As seen in Figure 8, for PNI = 1, the relative RY was near 1.0, while for PNI > 1 or PNI < 1, the relative RY decreased. The current study showed that inadequate and excessive P use lowers the relative RY. Only the optimal P application rate results in the maximum relative RY.

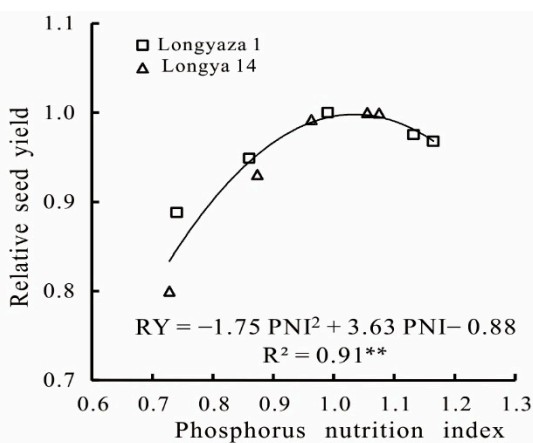

**Figure 8.** Relationship between relative seed yield (RY) and phosphorus nutrition index (PNI) for two flax cultivars. The PNI data were averaged over two years. ** Significance at *p* < 0.01 probability level.

## 4. Discussion

This study proposes the idea of the current Pc dilution phenomenon in the capsule of flax firstly. We have constructed a Pc dilution curve based on the CDM and provided a new way of diagnosing and regulating the P status of flax during the reproductive period in northwest China. Phosphate plays a pivotal role in the nexus of photosynthesis [27], energy conservation [28], and carbon metabolism [29] in higher plants and its application to agricultural soils is crucial to achieving optimum crop production [15]. However, excessive P fertilizer cannot ensure a significant increase in crop productivity, yet its abundant application can decrease crop yield and cause environmental damage. In recent years, there has been an increasing demand for simple, accurate, and stable tools for diagnosing the status of crop P, which can provide appropriate on-farm P management. With the advancement of the P dilution principle, the diagnosis model based on this theory has been developed and used to guide agricultural field production.

### 4.1. Comparison with Other Critical Phosphorus Dilution Curves

The Pc dilution curve based on DM (including aboveground DM, vines DM, tubers DM, etc.) has been previously established for different crop species and regions (Table 3). However, this study was the first to analyze the Pc dilution curve in the capsules of flax. Our results show that, as was observed for N [30], there is a dilution of P with increasing CDM. The Pc dilution curve, based on the CDM, was Pc = 2.84×CDM$^{-0.22}$, where 2.84 represents the Pc when CDM = 1 t, in this study.

Many differences in curve parameters are noticeable between our study and the earlier studies in other crops. In the current study, the value of parameter *a* was lower than the value of parameter a based on plant DM reported in younger timothy and older timothy [18], in potato [13,16,17,20], in wheat [14], and for winter wheat, maize, and rapeseed [15]. The discrepancy was related to genotype, circumstance, and growth stage differences. Firstly, the genotype of crops induces the differences between these curves. Genotype differences in critical P dilution curves have been reported in Switzerland with wheat, maize, and rapeseed [15]. The difference in critical P dilution curves within genotype is probably attributed to the general nature of crop species. Secondly, the growth environment of crops

resulted in discrepancies in the Pc dilution curves. This is because the difference from the growth environment may be attributed to differences in pedo-climatic conditions, water supply, soil properties, etc. The growth environment may have affected P uptake from the soil and partitioning by different crop apparatus. However, parameter a in this study was greater than those of parameter a based on tubers DM and vines DM reported in potatoes [17]. Corresponding to the tuber P dilution curve, the main reason may be that the P concentration of a potato tuber was smaller than the P level of a flax capsule, and to the vines' P dilution curve, other factors obviously influence the P dilution curve. Further research is required to be conducted to make a thorough inquiry.

**Table 3.** Comparison of the parameters of the critical phosphorus concentration (Pc) dilution curve for different regions and measured parameters of timothy, potato, wheat, maize, rapeseed, and flax.

| Crop | Region | $^e a$ | $^f b$ | Measured parameter | Reference |
|---|---|---|---|---|---|
| OT | Canada | 3.27 | 0.20 | DM [a] | Bélanger and Ziadi, 2008 [18] |
| YT | | 5.23 | 0.40 | | |
| Potato | Argentina | 3.92 | 0.30 | DM [b] | Zamuner et al., 2016 [13] |
| | Colombia | 5.23 | 0.19 | DM | Gómez et al., 2016 [16] |
| | Canada | 3.57 | 0.38 | DM | Nyiraneza et al., 2021 [17] |
| | | 2.58 | 0.20 | Vines DM | |
| | | 2.06 | 0.14 | Tubers DM | |
| | Brazil | 3.91 | 0.30 | DM | Soratto et al., 2020 [20] |
| Wheat | Canada | 4.94 | 0.49 | DM [c] | Bélanger et al., 2015 [14] |
| | Finland | 4.04 | 0.21 | DM | |
| | Canada | 3.62 | 0.23 | DM | |
| | China | 4.40 | 0.29 | DM | |
| WW | Switzerland | 4.44 | 0.41 | DM | Cadot et al., 2018 [15] |
| Maize | | 3.49 | 0.19 | | |
| Rapeseed | | 5.18 | 0.39 | | |
| Flax | China | 2.84 | 0.22 | CDM [d] | This article |

OT = older timothy; YT= younger timothy; WW = winter wheat; [a] DM, aboveground dry matter; [b] DM, vines + tubers dry matter; [c] DM, shoot dry matter; [d] CDM, capsule dry matter; [e] *a*, the critical plant P concentration in the capsule dry matter = 1 t ha$^{-1}$; [f] *b*, the decline in capsule phosphorus concentration with crop growth.

Parameter *b* describes the decline in CPC with crop growth and therefore depends on capsule P uptake relative to the CDM increase. The decline in CPC during reproductive growth can be attributed to the decrease in P concentration per unit CDM, similar to the N concentration decline per unit shoot DM in linseed [30]. The decrease in CDC is probably due to (i) a large photosynthetic product of other vegetative parts (stem and leaf) transferred to the seed for yield formation during the reproductive growth period [4,31], which causes the faster rate of CDM accumulation, and (ii) the P nutrition of pericarp and stalk was transferred to the seed for fatty acids and other nutrients formation, which results in a declining P concentration of pericarp and stalk during the reproductive growth period.

Compared with the DM model, the parameter b we established in this study was close to those curves proposed for older timothy in Canada [18], for potatoes in Colombia [16] and in Canada [17], for wheat in Finland [14], and for maize in Switzerland [15]. Whereas the difference existed between the parameter b in the current study and those of the curve developed for younger timothy [18], for potato [13,20] and potato [17], for wheat [14], and for wheat and rapeseed [15]. Those indicated that the rates of plant Pc dilution of the mentioned crops were faster than that of the capsule Pc dilution in flax. These differences were associated with the transfer of large amounts of P to the capsule to satisfy yield and fatty acid formation and a slower CPC decline in the capsule because the capsule was the epicenter of plant growth during the reproductive period.

*4.2. Diagnosis of Phosphorus Nutrition Status*

The present study validated the Pc curve based on the CDM with independent data. Our results indicated that this curve could accurately distinguish P-limiting and non-P-limiting treatments during the reproductive growth period of flax. For this purpose, the PNI value was calculated based on this Pc curve. This curve has been used to diagnose the P status of timothy [18], potato [13,16,17,20], wheat [14], maize, and rapeseed [15]. In general, those Pc curves were developed during the throughout growth period; only Nyiraneza et al. [17] proposed a tuber Pc dilution curve of potatoes during the reproductive period. Since the reproductive period was another peak of P uptake for flax [32], it is essential to develop a curve for diagnosing the P status of flax after anthesis.

In the study, PNI was the ratio between the measured capsule P concentration of flax and Pc. The PNI can effectively distinguish conditions of P deficiency, optimal levels, or P surplus of flax during the reproductive growth period. Confirmation of P sufficiency during this period of flax can help perform corrective measures, especially under irrigated conditions. Studies showed that foliar P fertilizer applications [33] could be supplied in-season to correct a P deficiency [34], improve photoassimilate production [35], and enhance dry matter distribution [36] and translocation from vegetation organs to seeds [4]; although, only a limited amount of P can be absorbed into the plant through foliage. Therefore, PNI can estimate phosphorus nutrition sufficiency, and this would represent a fast and cost-effective option. This new methodology offers the only reliable way to correct P nutrition during the critical period of flax growth and to ensure maximum yields without having to apply large doses of P at planting. This strategy could help to achieve much more efficient use of P fertilizer and to minimize the cost and environmental threat associated with applying P at high rates.

## 5. Conclusions

Based on the theory of Pc dilution, this study proposed a Pc dilution curve based on CDM for flax. The curve was described by the following equation, Pc = 2.84 CDM$^{-0.22}$, for the CDM of flax. The equation was derived from data obtained at five growth stages from two cultivars over five P fertilizer application rates and two growing seasons. The equation was validated using data from two separate cultivars under different sites and the same fertilizer conditions. The PNI values on different sampling dates were generally <1 under P-limiting and >1 under non-P-limiting conditions. There was a significant positive relationship between PNI and the P level across the five development stages, suggesting the Pc dilution curve can be used as a tool for diagnosing the P status of flax. In addition, the relationship between relative RY and PNI was well illustrated with a second-order polynomial equation, in which for PNI = 1, the relative RY was near 1.0, while for PNI > 1 or PNI < 1, the relative RY decreased. Hence, we conclude that the capsule Pc curve can be adopted as a practical diagnostic tool for effective P management during the reproductive growth period of flax in northwest China.

**Author Contributions:** Conceptualization, J.Z.; investigation, L.W., M.M.N., Z.L., Z.D., W.L., Y.Q. and W.Z.; project administration, Z.C., X.W., Y.Z. and G.W.; and data curation and original draft preparation, Y.X. and Y.L. All authors have read and agreed to the published version of the manuscript.

**Funding:** This research was funded by the Key Research and Development Projects of Gansu Academy of Agricultural Sciences (2021GAAS20), the National Natural Science Programs of China (31660368; 32060437), the Major Special Projects of Gansu Province (21ZD4NA022), and China Agriculture Research System of MOF and MARA (CARS-17-GW-04).

**Institutional Review Board Statement:** Not applicable.

**Informed Consent Statement:** Not applicable.

**Data Availability Statement:** The datasets used and/or analyzed during the current study are available from the corresponding author upon reasonable request.

**Conflicts of Interest:** The authors declare no conflict of interest.

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
