# Peer review of "Determination of Critical Phosphorus Dilution Curve Based on Capsule Dry Matter for Flax in Northwest China"

_agronomy, doi:10.3390/agronomy12112819_

Round 1
Reviewer 1 Report
Comments:
An interesting paper that needs some clarification for reader for publication. The authors refer to “capsule” dry matter throughout the paper but this is not defined nor made clear to reader what they are referring to: is it the flax boll? It reads in materials and methods as though the plant material they harvested, measured and analyzed was the entire above-ground material. Graphical depictions are well done. Define in the table 3 caption what “a” and “b” are in the table. The authors refer to foliar P application as a means to improve crop P nutrition in – season, but should also indicate that only a limited amount of P can be absorbed into the plant through foliage. Sometimes seems unnecessarily complex approaches are taken to analyze the data and tell the story.
Author Response
Answer: Thanks for offering some excellent, constructive suggestions for further improvement the manuscript!
1 We changed this sentence “Four experiments utilized a split-plot arrangement of treatments a randomized complete block design with three replicates” into “Randomized complete block design with three replicates was used for this study with a plot size of 20 m2 (4 m × 5 m).” And we added “Four P application rates (0, 40, 80, and 120 kg P2O5 ha-1) were applied to two flax cultivars (Table 1) and five P rates (0, 40, 80, 120, and 160 kg P2O5 ha-1) were applied to other two flax cultivars (Table 2).” In addition, we added “There were no obvious pest and disease attack during the growth of flax. Further crop management procedures followed common agricultural practices to ensure maximum potential productivity, i.e., no factor other than P was limiting.” in 2.2 Experimental design.
2 We improved the sentence “The cultivars were applied to the main plots, and P fertilizer was applied to subplots.” to “The cultivars were mainly planted of local agriculture department and farms.” And deleted “and P fertilizer was applied to subplots.”
3 We improved the sentence “40 mm irrigation was used on all plots of Dingxi before flowering on 21 June 2017 and 17 June 2018. The plots at the Yongdeng site received 40 mm irrigation prior to the flowering of flax on 15 June 2018 and 18 June 2019.” to “To ensure maximum potential productivity, 40 mm water was used to irrigate each plot before flowering of flax.”
4 We added the photograph as figure 1 on the capsule of flax in the manuscript.
5 We changed the sentence “The 30 plants of flax per plot (4 by 5 m) were collected during the reproductive stage (Days after sowing 86, 92, 98, 104, and 110 days) for determination of CDM and capsule P concentration (CPC). The CDM was determined by a forced-draft drying oven at 75°C until constant weight.” into “The 30 plants of flax per plot (2 by 1 m) were manually harvested during the re-productive stage (Days after sowing 86, 92, 98, 104, and 110 days). Flax capsules (Figure 1) were collected and were dried at 75°C until constant weight each date.”
6 Accordingly, we defined in the table 3 caption “a” represented the critical plant P concentration in the capsule dry matter =1 t ha−1 and “b” represented the decline in capsule phosphorus concentration with crop growth are in the table.
7 At the same time, we added the sentence “although, only a limited amount of P can be absorbed into the plant through foliage.” in the 4.2 Diagnosis of phosphorus nutrition status.
8 We added “In addition, the relationship between relative RY and PNI was well illustrated with a second-order polynomial equation, in which for PNI=1, the relative RY was near 1.0, while for PNI>1 or PNI<1, the relative RY decreased. Hence,” in the conclusions.
The content of added and revised were marked up using red font in the manuscript.
Thank you very much for your work concerning our manuscript.
Best wishes!
Yaping Xie, Jianping Zhang

Reviewer 2 Report
The idea of the current research is good, especially at the present time. The manuscript is well written as well. The presentation and discussion of the results were carried out perfectly. Just view notes,
1) Is there a possibility to show ANOVA table of the split-plot arrangement in RCBD?
2) What about the interaction between cultivars and growing seasons?
3) Is there genotypic variation among the four cultivars tested in relation to Pc, DM, Seed yield, etc?
4) If the authors (in Table 3) added the reference numbers according to the journal's format.

Author Response
The idea of the current research is good, especially at the present time. The manuscript is well written as well. The presentation and discussion of the results were carried out perfectly. Just view notes,
- Is there a possibility to show ANOVA table of the split-plot arrangement in RCBD?
Answer: Firstly, we very thanks for recognizing on our manuscript from reviewer. Secondly, we are very sorry! The study was not the split-plot arrangement. This is a slip in 2.2 Experimental design of the manuscript. It is randomized complete block design. We have amended in 2.2 Experimental design of the current manuscript. Moreover, we added the least significant difference (LSD) bars in the Figure 2 and 3. According the content of the manuscript, ANOVA table may not be added.
- What about the interaction between cultivars and growing seasons?
Answer: The interaction between cultivars and growing seasons were overlooked, due to cultivars have the same trend of change in each growing season.
- Is there genotypic variation among the four cultivars tested in relation to Pc, DM, Seed yield, etc?
Answer: There is genotypic variation among the four cultivars tested in relation to DM and seed yield.
4) If the authors (in Table 3) added the reference numbers according to the journal's format.
Answer: Thanks! We added the reference numbers according to the journal's format in Table 3.
The content of added and revised were marked up using red font in the manuscript.
Thank you very much for your work concerning our manuscript.
Best wishes!
Yaping Xie, Jianping Zhang